# Catalytic enantioselective nitrone cycloadditions enabling collective syntheses of indole alkaloids

Xiaochen Tian[1,2,4], Tengfei Xuan[1,2,4], Jingkun Gao[3,4], Xinyu Zhang[1], Tao Liu[1], Fengbiao Luo[1], Ruochen Pang[1], Pengcheng Shao[1], Yun-Fang Yang [3] ✉ & Yang Wang [1,2] ✉

Tetrahydro-β-carboline skeletons are prominent and ubiquitous in an extraordinary range of indole alkaloid natural products and pharmaceutical compounds. Powerful synthetic approaches for stereoselective synthesis of tetrahydro-β-carboline skeletons have immense impacts and have attracted enormous attention. Here, we outline a general chiral phosphoric acid catalyzed asymmetric 1,3-dipolar cycloaddition of 3,4-dihydro-β-carboline-2-oxide type nitrone that enables access to three types of chiral tetrahydro-β-carbolines bearing continuous multi-chiral centers and quaternary chiral centers. The method displays different endo/exo selectivity from traditional nitrone chemistry. The distinct power of this strategy has been illustrated by application to collective and enantiodivergent total syntheses of 40 tetrahydro-β-carboline-type indole alkaloid natural products with divergent stereochemistry and varied architectures.

The Natural products and synthetic compounds containing tetrahydro-β-carboline (THβC) skeletons are endowed with various biological activities[1,2]. THβCs as fundamental privileged heterocyclic frameworks are widely distributed in a large family of indole alkaloids, such as eburnamonine, arboricine, arbornamine (Fig. 1a). THβCs are also important backbones in a large variety of bioactive molecules and pharmaceutical compounds, such as yohimbine, tadalafil, and reserpine (Fig. 1a). Consequently, tetrahydro-β-carbolines covering a wide variety of structural types have represented attractive targets for synthesis, and have stimulated the development of efficient approaches and strategies to build up such architectures[3]. Synthetic methodologies especially in asymmetric manner are of considerable interest and in great demand and have been constant pursuit for chemists. Among the reported methods, one time-honored method for the synthesis of chiral THβCs is asymmetric Pictet–Spengler (PS) reaction of tryptamines or tryptamine derivatives with aldehydes or ketones, which presents the most direct and straightforward strategy. Since the pioneering enantioselective catalytic PS transformation reported by List in 2006[4], a large number of asymmetric variants of this reaction have been developed (Fig. 1b-a)[5–11]. In addition, other synthetic methods developed thus far such as asymmetric hydrogenation of imines[12–16] or iminium salts[17,18] bearing dihydro-β-carboline (DHβC) moiety (Fig. 1b-b), and catalytic enantioselective addition of nucleophiles to cyclic imines or iminium electrophiles[19–21] (Fig. 1b-c) have been shown to be elegant alternative methods toward optically pure THβCs. Some advances in enantioselective oxidative C–H functionalization of N-carbamoyl THβCs (Fig. 1b-d)[22–24], aza-Diels-Alder reaction of 3-vinylindoles with ketimines followed by migration (Fig. 1b-e)[25], intramolecular cyclization of N-allenamides (Fig. 1b-f)[26], [3 + 3] annulation of 2-vinylindoles and aziridines (Fig. 1b-g)[27], and multi-step synthesis involving carbene insertion/intramolecular aza-Michael reaction/hydrogenation[28] (Fig. 1b-h) have provided powerful

[1]Molecular Synthesis Center & Key Laboratory of Marine Drugs, Chinese Ministry of Education, School of Medicine and Pharmacy, Ocean University of China, Qingdao, China. [2]Laboratory for Marine Drugs and Bioproducts, Qingdao Marine Science and Technology Center, Qingdao, China. [3]State Key Laboratory Breeding Base of Green Chemistry-Synthesis Technology, Key Laboratory of Green Chemistry-Synthesis Technology of Zhejiang Province, College of Chemical Engineering, Zhejiang University of Technology, Hangzhou, Zhejiang, China. [4]These authors contributed equally: Xiaochen Tian, Tengfei Xuan, Jingkun Gao. ✉e-mail: yangyf@zjut.edu.cn; wangyang@ouc.edu.cn

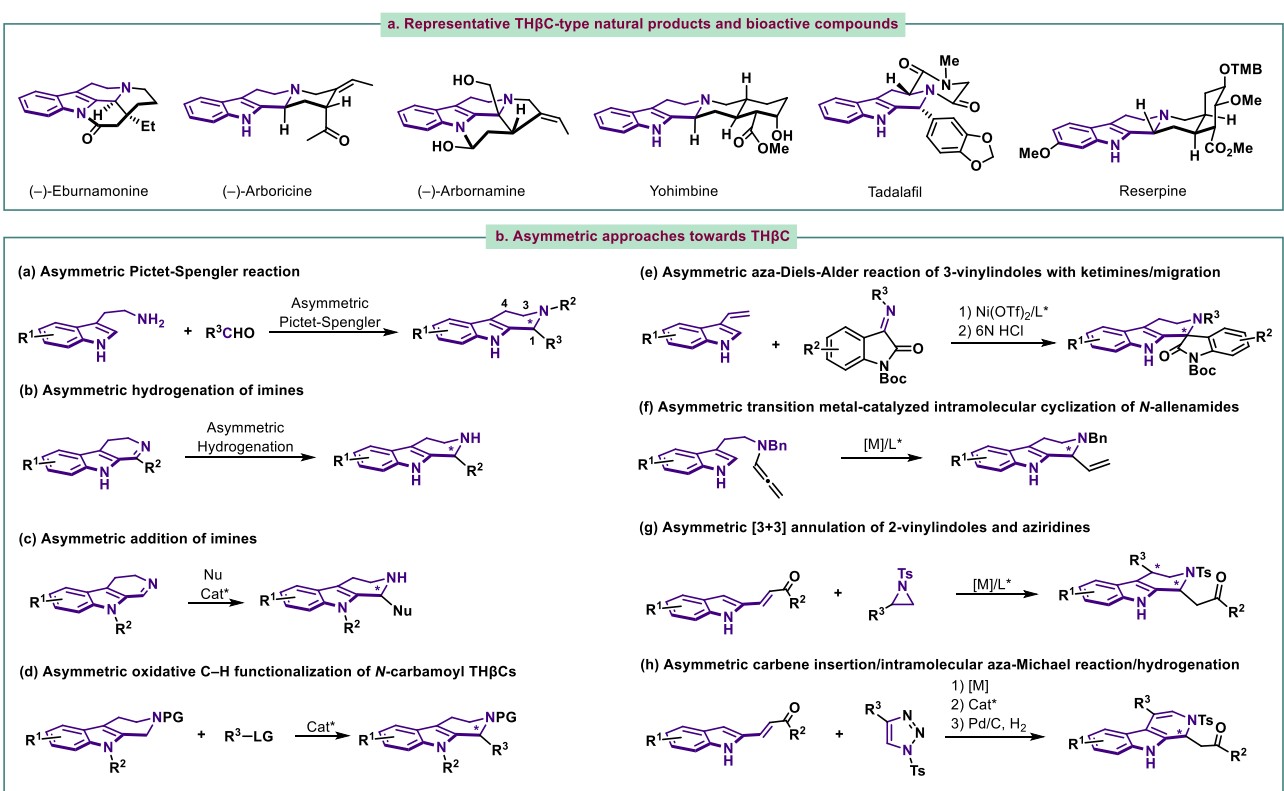

**Fig. 1 | Relevance of THβCs and state of the art in catalytic asymmetric syntheses of THβCs. a** Representative THβC-type natural products and bioactive compounds. **b** Previous asymmetric approaches towards THβCs.

strategies for asymmetric synthesis of C1-substituted THβCs. Recently, intramolecular cyclization such as cyclization of 2-aminoaryl alkynones[29], allylic alkylation of 2-indolyl allyl carbonates[30–32], and allylic dearomatization/migration of 3-indolyl allyl carbonates[33] has displayed the forceful ability in the construction of C4-substituted chiral THβCs.

Despite the considerable progress for enantioselective synthesis of THβCs, existing methods commonly face significant challenges that remain to be solved. First is how to synthesize THβC derivatives with continuous multi-chiral centers in one-step, especially the continuous stereocenters adjacent to C1 position. Till now, most methods focus on the step-by-step construction of continuous stereocenters initiated by first building chiral enter at C1 position. There have been only several examples reported to access continuous stereocenters in one step[15,34–36]. Second is how to synthesize 1,1-disubstituted THβCs possessing C1-chiral quaternary carbon center enantioselectively. Enantioselective formation of quaternary stereocenters is a longstanding crucial challenge in THβCs synthesis. To date, most catalytic asymmetric methods towards 1,1-disubstituted THβCs focused on PS-type reactions[34,37–47]. Asymmetric alkylation of 1-cyano substituted THβCs[48] as well as Friedel–Crafts reaction between isoindolo-β-carboline-derived hydroxylactam and indole have been reported to furnish 1,1-disubstituted THβCs[49]. Third is how to illustrate the utility of the developed methods in the synthesis of natural products and biologically active molecules. Efficient collective assembly of complex natural products with architectural and stereochemical divergence from a general method remains highly desirable. Although THβCs are core structures of natural products and integral parts of bioactive compounds, aside from PS reaction and Noyori asymmetric hydrogenation of imine or iminium, a few methodologies have been widely applied to the total synthesis of THβC-type natural products and bioactive compounds[15,24,25,46,50–54]. The development of a general method for

collective total syntheses of diverse THβC-type indole alkaloid natural products and bioactive compounds remains a formidable challenge.

To address these challenges, we hypothesized that a possible solution might arise from investigation of unified platform starting materials for their ability to construct quaternary stereocenters and undergo further transformation for a large collection of natural product syntheses. We considered that 3,4-dihydro-β-carboline-2-oxide (Fig. 2), a type of nitrone containing DHβC skeleton, represented an attractive platform molecule to synthesize THβC-type structural motif. Taking advantage of nitrone chemistry, 3,4-dihydro-β-carboline-2-oxide has been reported to be reactive in 1,3-dipolar cycloaddition with dipolarphiles[55–61]. However, the 3,4-dihydro-β-carboline-2-oxide chemistry was unexplored in enantioselective form. Towards this end, we wished to develop the asymmetric conversion of 3,4-dihydro-β-carboline-2-oxides and explore their application in natural product syntheses. Herein, we disclosed catalytic enantioselective 1,3-dipolar cycloaddition of 3,4-dihydro-β-carboline-2-oxide with vinyl ether, providing a highly efficient protocol toward three types of chiral tetrahydro-β-carboline fused isoxazolidines containing two stereocenters, three continuous stereocenters, and quaternary stereocenters through varying easily accessible vinyl ether substrates. This work presented a strategy for enantioselective synthesis of THβCs. More importantly, the *endo/exo* selectivity of our 1,3-dipolar cycloaddition differed from traditional nitrone chemistry[62,63]. The presence of nitrone group in 3,4-dihydro-β-carboline-2-oxide can controllably and easily lead into functionalization for further transformation. The resultant formation of N–O bond of isoxazolidine conferred structural advantages that could be cleaved by assorted alkyl halides to induce additional functionality and complexity that would be difficult to be achieved from traditional approaches. This protocol enabled access to 40 THβC-type indole alkaloid natural products belonging to five families and

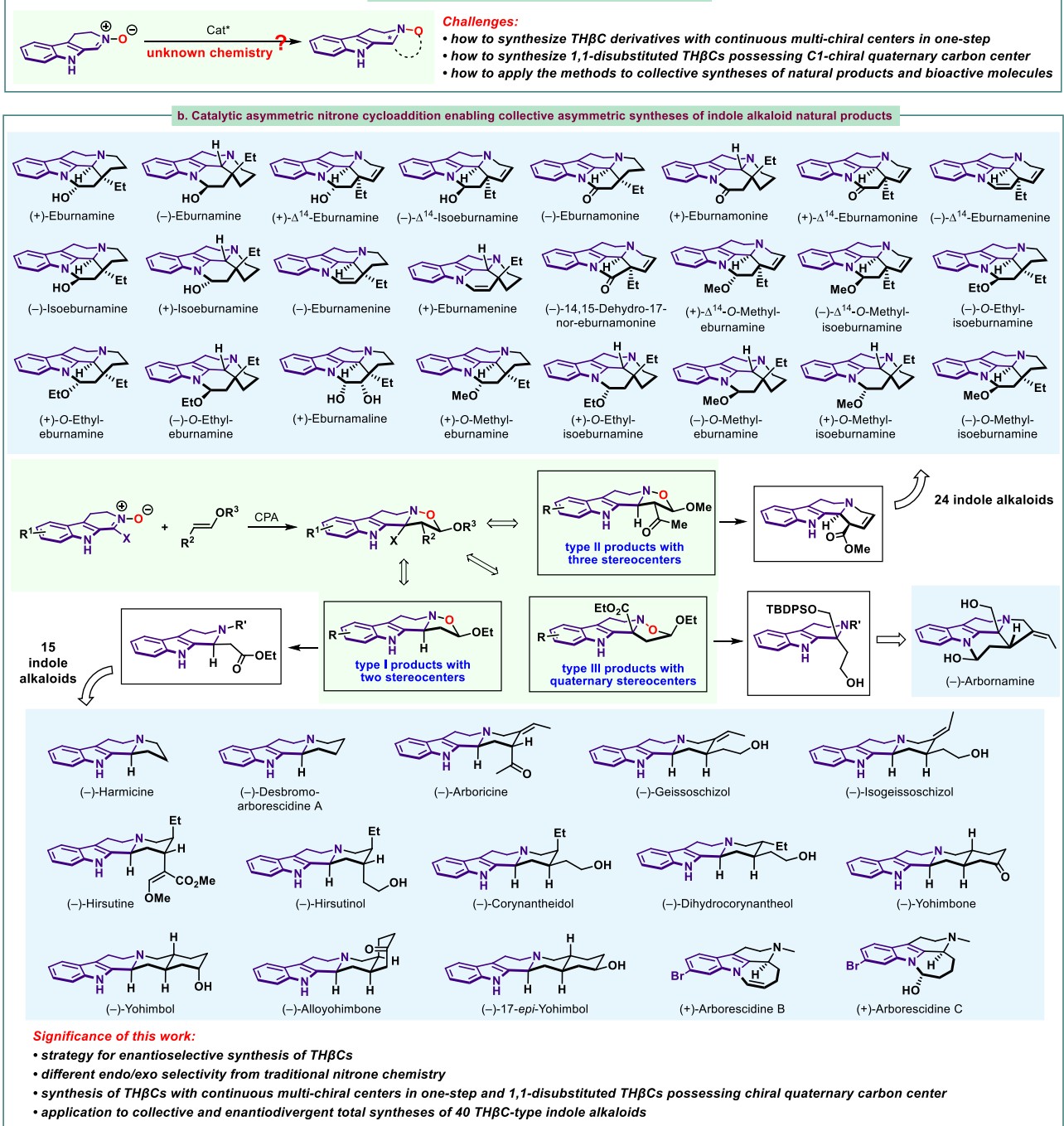

**Fig. 2 | Catalytic asymmetric synthesis of THβCs from nitrone. a** Development of the asymmetric conversion of 3,4-dihydro-β-carboline-2-oxides. **b** Our design of unified platform starting materials to construct various THβCs with varied architectures for collective total syntheses of natural products.

enantiodivergent total syntheses of eburnane-alkaloids, therefore rendered collective syntheses of natural products.

## Results

To test our hypothesis, the preliminary studies began with the investigation of chiral phosphoric acid[64–74] catalyzed 1,3-dipolar cycloaddition of 3,4-dihydro-β-carboline-2-oxide with vinyl ether (Table 1). 3,4-Dihydro-β-carboline-2-oxide **1a** and vinyl ether **2a** were chosen as initial substrates. The reaction catalyzed by BINOL-derived chiral phosphoric acid **4a** was first conducted in DCM at −20 °C. The desired tetrahydro-β-carboline fused isoxazolidine **3a** was obtained in 37% yield with 5:1 dr and 50% ee (entry 1). Then, different solvents were

screened (entries 1–6), and CHCl₃ (entry 2) was found to be the best solvent, affording **3a** with moderate diastereoselectivity (7:1 dr) and enantioselectivity (73% ee). Molecular sieves were proved to be beneficial for the improvement of enantioselectivity (entries 7–9), and the ee value was improved to 81% with the addition of 3 Å MS (entry 7). Encouraged by this promising result, various chiral phosphoric acids were screened. To our delight, the yield of **3a** was dramatically enhanced without deterioration of diastereoselectivity and enantioselectivity when BINOL-derived CPA **4b** was used as catalyst (entry 10). CPAs **4e** derived from SPINOL (entry 13) and **4j** derived from H₈-BINOL (entry 14) couldn't give higher enantiomeric excess. Thus, extensive evaluation of CPA catalysts confirmed that the optimal catalyst was **4b**.

**Table 1 | Screening the reaction conditions**

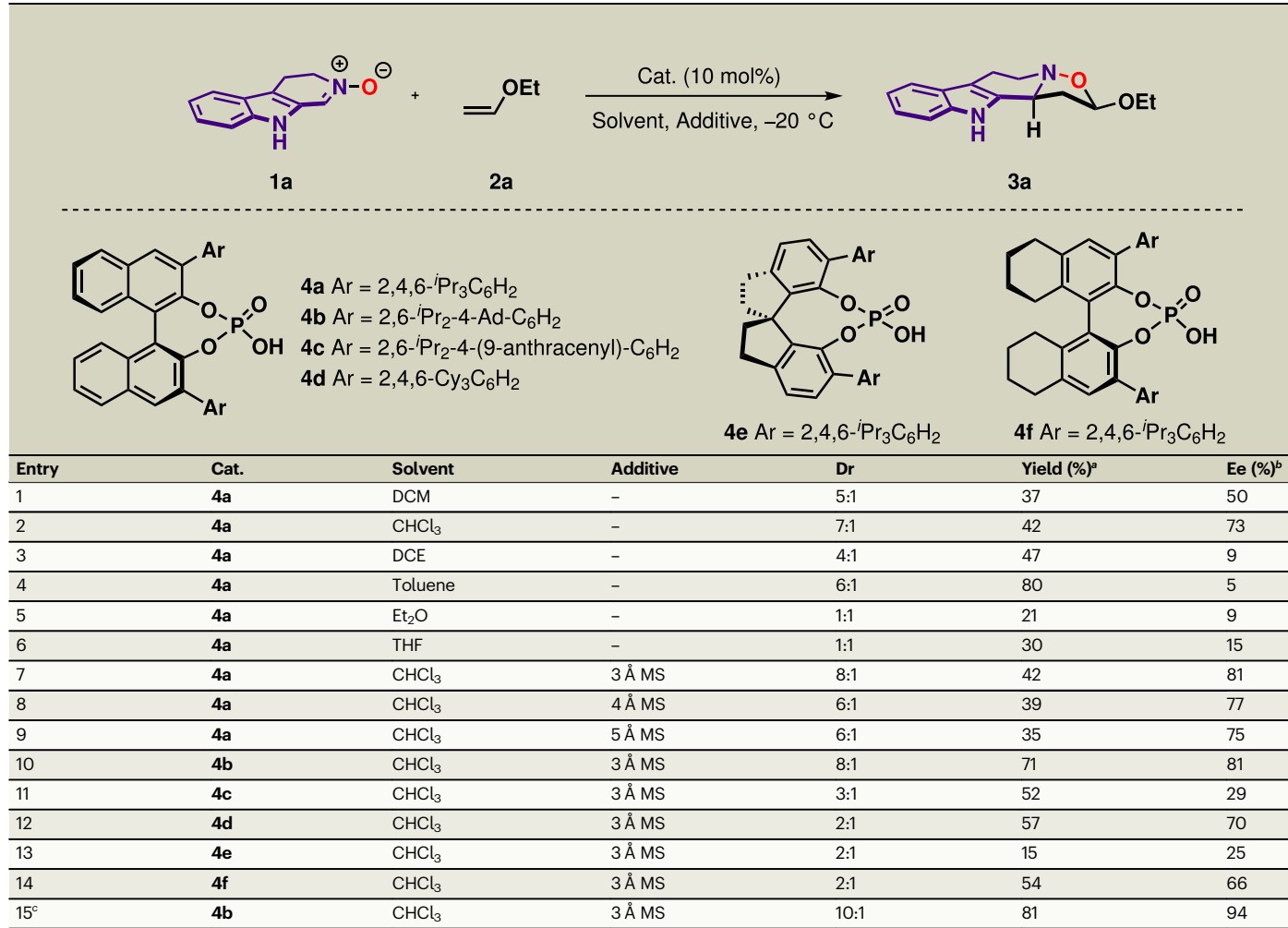

| Entry | Cat. | Solvent | Additive | Dr | Yield (%)[a] | Ee (%)[b] |
|---|---|---|---|---|---|---|
| 1 | 4a | DCM | – | 5:1 | 37 | 50 |
| 2 | 4a | CHCl₃ | – | 7:1 | 42 | 73 |
| 3 | 4a | DCE | – | 4:1 | 47 | 9 |
| 4 | 4a | Toluene | – | 6:1 | 80 | 5 |
| 5 | 4a | Et₂O | – | 1:1 | 21 | 9 |
| 6 | 4a | THF | – | 1:1 | 30 | 15 |
| 7 | 4a | CHCl₃ | 3 Å MS | 8:1 | 42 | 81 |
| 8 | 4a | CHCl₃ | 4 Å MS | 6:1 | 39 | 77 |
| 9 | 4a | CHCl₃ | 5 Å MS | 6:1 | 35 | 75 |
| 10 | 4b | CHCl₃ | 3 Å MS | 8:1 | 71 | 81 |
| 11 | 4c | CHCl₃ | 3 Å MS | 3:1 | 52 | 29 |
| 12 | 4d | CHCl₃ | 3 Å MS | 2:1 | 57 | 70 |
| 13 | 4e | CHCl₃ | 3 Å MS | 2:1 | 15 | 25 |
| 14 | 4f | CHCl₃ | 3 Å MS | 2:1 | 54 | 66 |
| 15[c] | 4b | CHCl₃ | 3 Å MS | 10:1 | 81 | 94 |
| 16[d] | 4b | CHCl₃ | 3 Å MS | >19:1 | 85 | 98 |

Reaction conditions: **1a** (0.20 mmol), **2a** (0.40 mmol), and catalyst **4** (0.02 mmol) in solvent (2.0 mL) at –20 °C.
[a]Isolated yield, dr were determined via 1H NMR of the crude products.
[b]The ee values were determined by chiral HPLC analysis.
[c]The reaction was conducted at –40 °C.
[d]The reaction was conducted at –60 °C.

Then, it is indicated that lower reaction temperature was better for the diastereoselectivity and enantioselectivity. Overall, systematic evaluation of the reaction variables such as solvent, additive, catalyst, and reaction temperature identified the optimal reaction condition shown in entry 16, which gave **3a** in 85% yield with >19:1 dr and 98% ee.

With the optimized reaction conditions in hand, we sought to apply this catalytic enantioselective protocol to the synthesis of an array of chiral functionalized tetrahydro-β-carboline fused isoxazolidines. As shown in Fig. 3, we first evaluated the substrates scope of 3,4-dihydro-β-carboline-2-oxide **1** in the reaction with vinyl ether **2a**. It is revealed that 3,4-dihydro-β-carboline-2-oxide substrates reacted smoothly in this cycloaddition reaction. The 4-, 5-, 6- or 7-positions of indole core in **1** could be substituted by various electron-rich and electron-deficient groups, affording the desired tetrahydro-β-carboline fused isoxazolidine products in good yields with high diastereoselectivities and enantioselectivities (**3a**–**3m**). 3,4-Dihydro-β-carboline-2-oxide substrates bearing increased steric hindrance around 7-substitution patterns of indole ring exhibit excellent enantioselectivities as well. The absolute configuration of **3g** was confirmed by X-ray diffraction analysis and those of other products were assigned accordingly. We also demonstrated the scalability of this protocol by a 5 mmol scale experiment of **1a** with **2a** under standard reaction

condition. The gram-scale reaction smoothly provided **3a** in 82% yield with >19:1 dr and 98% ee. Cyclic vinyl ether was compatible in this reaction as well, and the transformation proceeded well when using DCE as solvent, furnishing **3n** in 78% yield with 90% ee. We next turned our attention to the use of other type of alkene to investigate the substrate scope for establishing three continuous stereocenters. When the alkene was changed to (*E*)-4-methoxy-3-buten-2-one **2b**, 3,4-dihydro-β-carboline 2-oxide bearing electron-rich and electron-deficient groups on the indole core were well tolerated regardless of the steric effect. In these cases, the corresponding tetrahydro-β-carboline fused isoxazolidine products (**5a**–**5t**) were afforded in high yields with excellent diastereoselectivities and enantioselectivities. It is noteworthy that the enantiodivergent cycloaddition between **1a** and **2b** was performed as well, enantiomer **ent-5a** was obtained in 91% yield with 93% ee when using (*R*)-**4b** as catalyst. The absolute configuration of **5d** was confirmed by X-ray diffraction analysis and those of other products were assigned accordingly. In an effort to broaden the substrates scope, we sought to use different nitrone substrates that may provide access to other THβC skeletal frameworks, which were difficult to be accessed by previous synthetic protocols. Enantioselective synthesis of 1,1-disubstituted THβCs is one of the most difficult tasks in THβCs synthesis. To this end, we went on to examine whether the developed

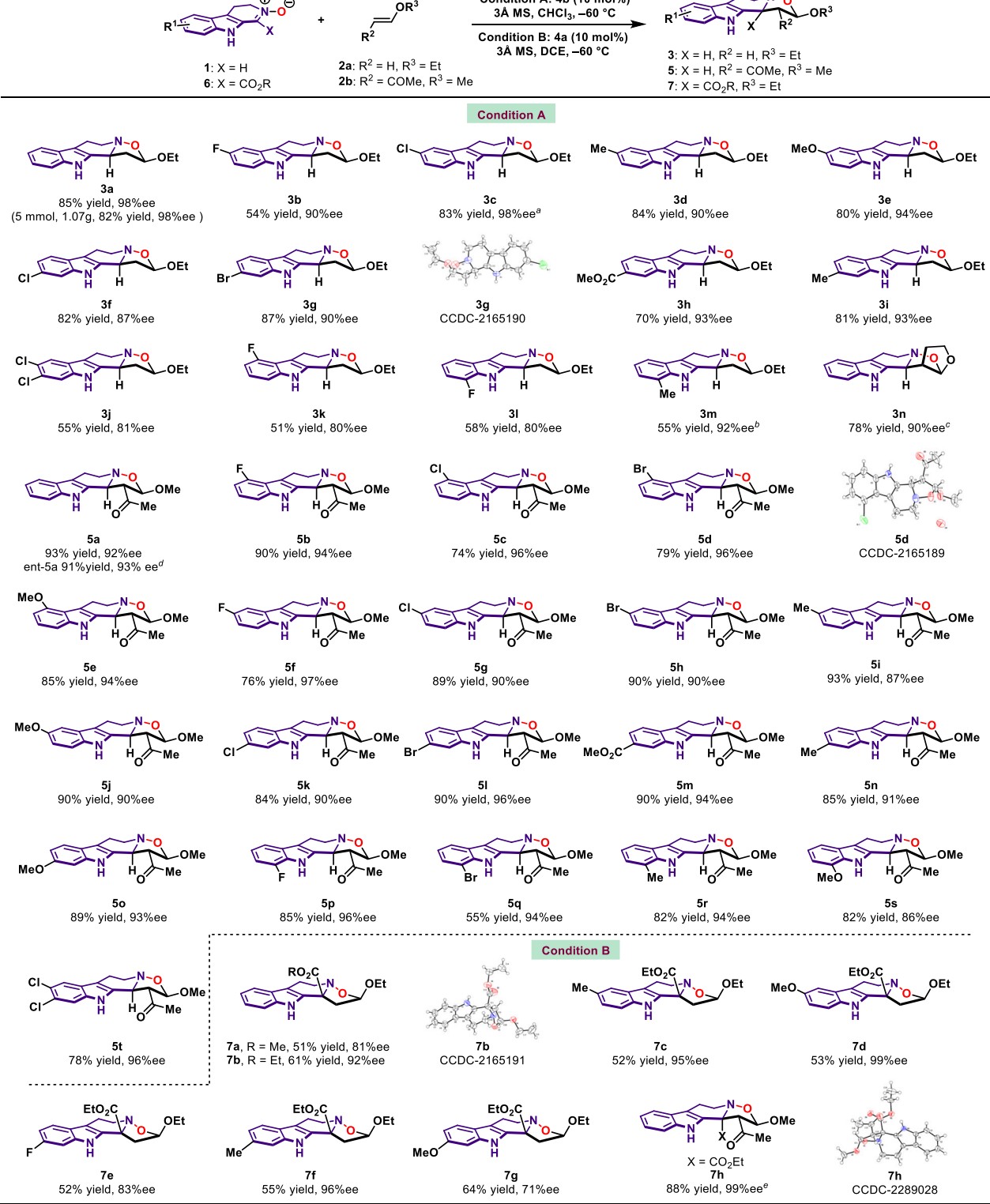

**Fig. 3 | Substrate scope of nitrones and vinyl ethers.** Reaction condition **A**: 1 (0.20 mmol), **2a** or **2b** (0.40 mmol), catalyst (0.02 mmol), and 3 Å MS (300 mg) in CHCl₃ (2.0 mL) at −60 °C. Reaction condition **B**: **6** (0.20 mmol), **2a** (0.40 mmol), catalyst (0.02 mmol), and 3 Å MS (300 mg) in DCE (2.0 mL) at −60 °C. Isolated yield.

d.r. >19:1. The ee values were determined by chiral HPLC analysis. $^a$ 6-$^n$Oct-**4a** was used. $^b$ DCM was used. $^c$ DCE was used. $^d$ (*R*)-**4b** was used as catalyst. $^e$ Toluene as solvent at −20 °C.

method could be extended to the construction of chiral 1,1-disubstituted THβCs with sterically congested C1-quaternary stereocenter. When nitrone **6** was employed, the transformation showed excellent performance on this type of substrates bearing a great variety of functional groups, leading to efficient construction of chiral 1,1-

disubstituted THβCs (**7a–7g**) in high efficiency with excellent stereoselectivities. The absolute configuration of **7b** was confirmed by X-ray diffraction analysis and those of other products were assigned accordingly. Moreover, we further showed the ability of our developed asymmetric cycloaddition method in the construction of

chiral 1,1-disubstituted THβCs with three continuous stereocenters containing one quaternary stereocenter, and the reaction proceeded smoothly, leading to the formation of **7h** in 88% yield with 99% ee.

In previous reported CPA-catalyzed nitrone chemistry, when nitrone underwent 1,3-dipolar cycloaddition with vinyl ether, Z-type nitrone delivered endo products[62], while E-type nitrone delivered exo products[63]. Due to the H-bonding interaction between CPA and nitrone oxygen atom, the traditional endo/exo selectivity could be rationalized by the steric repulsion between the substrates. Interestingly, the endo/exo selectivity of our 1,3-dipolar cycloaddition differed from previous works. In our work, opposite results were observed. Z-nitrones gave exo products and E-nitrones gave endo products. To further understand the reaction mechanism, the density functional theory (DFT) calculations were performed to explore the origin of the endo- or exo-selectivity. The FMO analysis provides a clear mechanistic model for these dipolar cycloaddition reactions involving various combinations of dipoles and dipolarophiles. The most stable transition state structures for endo- and exo-cycloaddition reactions are illustrated in Fig. 4a and Supplementary Fig. 6, aligning well with experimental selectivity. In cycloaddition reactions between 1,3-dipole **1a** and vinyl ethers **2a**, it is rationalized that H-bonding between nitrone oxygen atom and CPA resulted in a repulsive effect between the ethoxy group and the catalyst in exo type **TS1b**, leading to 4.3 kcal/mol higher free energy of **TS1b-exo** than that of **TS1a-endo**. The much smaller steric hindrance of endo approach **TS1a** allowed vinyl ether to react with nitrone via favored endo selective way. However, the DFT calculations revealed that the reaction of nitrone **1a** with vinyl ether **2b** exhibited different H-bonding interaction. In the **TS2**, CPA acted as bifunctional catalyst activating both nitrone and vinyl ether via H-bonding interaction between indole N–H group and vinyl ether alkoxy group. The steric repulsion between methoxy group and CPA made exo selective **TS2b** disfavored than endo selective **TS2a**. Ester group substituted Z-type nitrone **6** exhibited different endo/exo selectivity. When Z-type nitrone **6b** reacted with vinyl ether **2a**, the exo approach via **TS3b** was more favored than endo approach via **TS3a**. The steric repulsion between the ethoxy group of vinyl ether and the catalyst in endo selective **TS3a** leaded to the result that favored exo cycloaddition product via **TS3b** was obtained. When ester-substituted Z-type nitrone **6b** reacted with vinyl ether **2b**, the steric repulsion between the methoxy group of vinyl ether and the catalyst in exo selective **TS4b** was considered to be a major steric hindrance factor. Thus, an endo type product via **TS4a** was observed.

DFT calculations demonstrated that the hydrogen bonding interaction between chiral phosphoric acid catalysts and substrates facilitated the 1,3-dipolar cycloaddition reactions. In contrast, the uncatalyzed 1,3-dipolar cycloaddition reactions exhibited higher energy barriers compared to their chiral phosphoric acid-catalyzed counterparts (Supplementary Figs. 7 and 8). Notably, the dipolar cycloaddition transition states predominantly followed a concerted pathway[75], where C–C and C–O bonds formed simultaneously within a single transition state. Interestingly, for the cycloaddition reaction involving **1a** and **2b** with catalyst **4b**, a favorable stepwise pathway was identified, with C–O bond formation occurring first via **TS2a-endo-I**, followed by the C–C bond formation via **TS2a-endo-II**, which was the rate-determining step (Supplementary Fig. 14). We postulated that the favorable H-bonding interaction between **4b** and **2b** stabilized the zwitterionic intermediate, allowing for the localization of the stepwise transition states. On the contrary, the exo-cycloaddition via the concerted transition state **TS2b-exo** required a higher barrier than **TS2a-endo-II** by 6.0 kcal/mol. In reactions between ester-substituted Z-type nitrone **6b** and vinyl ether, **2b**, a favorable n → π* interaction[76] (from the lone pair of ethoxyl group O of **2b** to the π* orbital of the carbonyl group of **6b**) existed in the transition state **TS3b-exo**, which might be the origin of the exo-selectivity.

The additional mechanistic models exhibiting higher energies compared to those depicted in Fig. 4 were presented in Supplementary

Figs. 9–13. According to frontier molecular orbital (FMO) analysis (Fig. 4b), in cycloaddition reactions between 1,3-dipole **1a** and vinyl ethers **2a**, the primary FMO interaction occurred between the HOMO of dipolarophile **2a** and the LUMO of 1,3-dipole **1a**. Hence, the phosphoric acid preferred to interact with dipole **1a** to lower its LUMO. This interaction pattern was also observed in reactions between **6b** and **2a**. In cycloaddition reactions between **6b** and **2b**, the energy gap between the HOMO of **6b** and the LUMO of **2b** (10.28 eV) closely matched that between the HOMO of **2b** and the LUMO of **6b** (10.75 eV). The DFT-computed results suggested that the phosphoric acid similarly preferred to interact with dipole **6b** to lower its LUMO. However, for cycloaddition reactions between **1a** and **2b**, the primary FMO interaction occurred between the HOMO of 1,3-dipole **1a** and the LUMO of dipolarophile **2b**, differing from that of the previous cases. Here, the phosphoric acid catalyst **4b** tended to activate **2b** by forming hydrogen bonding interactions with the acetyl group of **2b**.

As mentioned before, the presence of nitrone group in 3,4-dihydro-β-carboline-2-oxide and the resultant N–O bond in the adducts could provide handles for further functionalization. To showcase the synthetic utility and versatility of our developed method, we commenced our study with postfunctionalization of tetrahydro-β-carboline fused isoxazolidine products (Fig. 5). The N–O bond of isoxazolidine conferred structural advantages that could be cleaved by different alkyl halides to induce functionality and complexity, thus unlocking access to THβC scaffolds bearing different substituents. With this aim in mind, a range of alkyl halides with various functionalities were subjected to **3a**. By forming quaternary ammonium salt intermediate followed with DABCO-promoted ring opening, the THβCs **8a**–**8d** could be afforded in good yields with maintained enantiopurity (Fig. 5, equation 1)[77]. **3a** was further derivatized through N–O bond cleavage via SmI₂/MeOH reduction and Boc protection, delivering primary alcohol **9** in 59% yield for two steps with 91% ee (Fig. 5, equation 2)[78]. So far, the enantioenriched THβC product **3a** has been readily transformed into a diversity of chiral THβC-type compounds with additional functionality and complexity. Therefore, to further showcase the synthetic versatility of our developed method, we anticipated the developed catalytic asymmetric 1,3-dipolar cycloaddition to provide a general platform for the collective syntheses of THβC-type indole alkaloids. The three types of chiral cycloaddition products **3**, **5**, and **7** with two stereocenters, three continuous stereocenters, and quaternary stereocenters can be further transformed, resulting in important building blocks and advanced intermediates for the synthesis of natural products and their analogs. While notable, all the synthetic transformations shown in Fig. 5 were used in the following total syntheses application. These synthetic applications will be discussed one by one.

As shown in Fig. 6, N–O bond cleavage of **3a** by using BnBr and Boc protection of indole N–H gave ester **10** in 95% yield by one-pot process. Subsequent DIBAL-H reduction followed by reaction with Wittig reagent and hydrolysis with HCO₂H/DCM afforded aldehyde **11** in 50% yield over two steps, which underwent a sequential deprotection/reduction to obtain compound **12** in 76% yield. Removal of Boc group gave natural product (−)-harmicine **13**[79] in 91% yield, which exhibited antileishmanial and antinociceptive activities[80] (Fig. 6a). When allyl bromide was used in the N–O bond cleavage and Boc protection process, the corresponding product **14** was obtained in 75% yield. Reduction of ester group and subsequent Wittig olefination afforded compound **15** in 75% yield over two steps. Then Grubbs II catalyst catalyzed RCM reaction proceeded smoothly to produce tetracyclic compound **16** in 80% yield. Hydrogenation using H₂ over Pd/C followed by deprotection of Boc provided (−)-desbromoarborescidine A **17**[81] in 88% yield over two steps, which exhibited antiproliferative activity[82] (Fig. 6b).

Next, we attempted the total synthesis of corynanthe as well as deplancheine family indole-alkaloids. When (Z)-1-bromo-2-iodo-2-butene

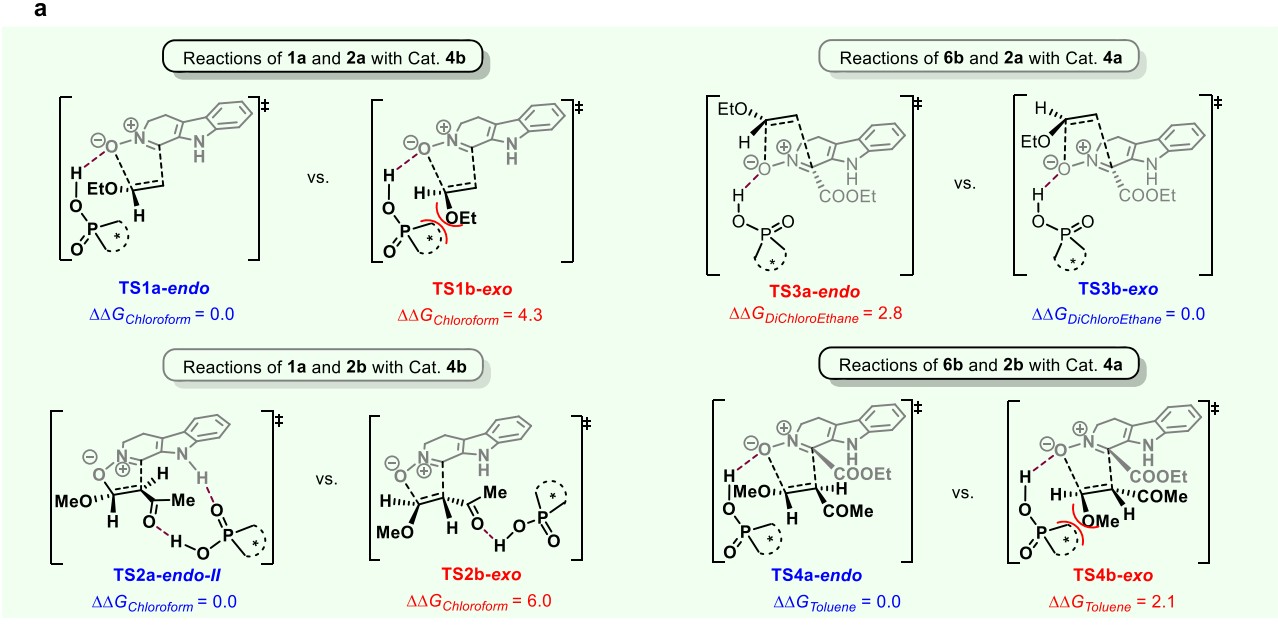

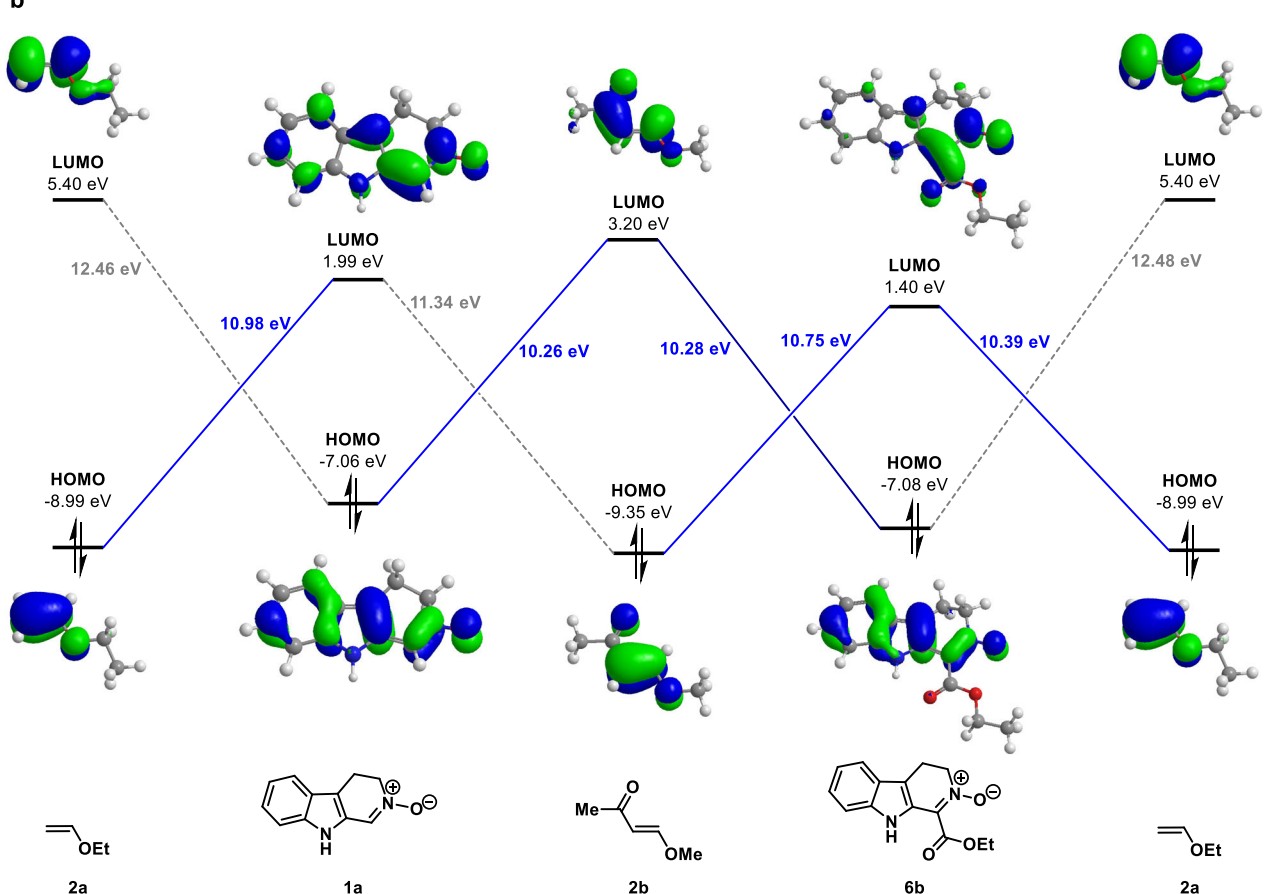

**Fig. 4 | Investigation of the mechanism. a** The key transition states of 1,3-dipolar cycloaddition reactions. Gibbs free energy (in kcal/mol) obtained at the level of SMD(solvent)-B3LYP-D3/def2TZVP//B3LYP-D3/6-31G*. In accordance with experimental conditions, chloroform was utilized as the solvent for reactions involving **1a** and **2a/2b** with catalyst **4b**. Dichloroethane was employed as the solvent for reactions of **6b** and **2a** with catalyst **4a**. Additionally, toluene was the chosen solvent for reactions of **6b** and **2b** with catalyst **4a**. **b** FMO diagram for the 1,3-dipolar cycloaddition reactions. HF/6-31G*//B3LYP-D3/6-31G*-computed orbital energies in eV are shown.

as ring opening reagent was subjected to **3a**, vinyl iodide **18** could be obtained in 82% yield. DIBAL-H reduction of ester afforded alcohol **19** in 76% yield. Mesylation of the primary alcohol and cyanation of mesylate led to the formation of cyano derivative **20**[83]. Then methyl Grignard

reagent was used for nucleophilic addition to access ketone **21** in 72% yield. Natural product (−)-arboricine **22**[84] was afforded in 64% yield by Pd-catalyzed intramolecular cyclization and silica gel promoted deprotection process, which has been reported to reverse multidrug resistance in

**Fig. 5 | Post-transformation.**

vincristine-resistant KB (VJ300) cells[84]. Controlled reduction of **18** to aldehyde followed by Wittig reaction provided ester **23** in 55% yield over two steps. Removal of Boc group by TFA and Ni-promoted intramolecular reductive cyclization gave intermediate **24** and diastereoisomer **25**. Reduction of **24** by Crabtree's catalyst gave single product **26**. Reduction of ester in **26** generated (−)-hirsutinol **27** in 81% yield. Condensation of **26** with methyl formate followed by methylation finished the synthesis of (−)-hirsutine **29**. Alternatively, LiAlH$_4$ reduction of **25** furnished (−)-geissoschizol **30**[85] in 84% yield. Moreover, **25** was reduced by PtO$_2$/H$_2$ to produce **31, 32** and **33**. These compounds were reduced by LiAlH$_4$ separately to obtain (−)-corynantheidol **34**[86,87], (−)-dihydrocorynantheol **35**[88], and (−)-isogeissoschizol **36**[89]. Therefore, seven corynanthe family indole alkaloids were synthesized from the same starting material **3a** and ring opening reagent alkyl bromide (Fig. 6c).

Yohimbine as pharmacological probe for the study of α2-adrenoceptor is a prescription drug for the treatment of impotence. Yohimbine is also used for other diseases. The realization of our synthetic method to yohimbine-type alkaloids is our next subject. As illustrated in Fig. 6d, when 2,3-dibromoprop-1-ene was used in N–O bond cleavage step, similar intermediate **38** could be isolated. Under AIBN and $^n$Bu$_3$SnH conditions, radical cyclization of **38** produced desired cyclization product **39** in 65% yield. Boc protection of **39** followed by DIBAL-H reduction of ester group to aldehyde gave **40**. Then, nucleophilic addition with allylic Grignard reagent and the following Hoveyda–Grubbs II catalyzed RCM reaction afforded **41** and diastereoisomer **42**. Double bond of **41** was reduced to obtain **43** and diastereoisomer **47**. Removal of Boc group in **43** produced (-)-17-*epi*-yohimbol **44** in 82% yield. Double bond reduction of **42** followed by deprotection of Boc generated (−)-yohimbol **45**[90]. (−)-Yohimbol **45** underwent efficient Pfitzner-Moffatt oxidation of hydroxy group, affording (−)-yohimbone **46** in 60% yield. Oxidation of hydroxy group and deprotection of Boc in **47** completed the total synthesis of (−)-alloyohimbone **48**. Therefore, four yohimbine-type indole alkaloids were synthesized from the same starting material **3a** and ring opening reagent alkyl bromide (Fig. 6d). When 6-Br substituted product **3g** was used as starting material, MeI was used in the N–O bond cleavage process to obtain ester **49**. Sequential reduction of ester, Wittig reaction, and reduction of double bond generated common intermediate **50**. Reduction of ester in **50** under different conditions could produce antiproliferative (+)-arborescidine B **51**[82] and (+)-arborescidine C **52**[91] separately (Fig. 6e).

We have established a general straightforward sequence based on our asymmetric 1, 3-dipolar cycloaddition for successful achievement of collective total syntheses of 15 THβC-type indole alkaloid natural products, relying on N–O bond cleavage of **3** by various alkyl halides. Then, we turned our attention to explore the synthetic application of another type of product **5a** with three continuous stereocenters in

eburnane-type indole alkaloids synthesis. Eburnane-type alkaloids, isolated from the plants of genus *Kopsia*, possessed potent bioactivities and have attracted enormous attention from synthetic chemists and medicinal chemists. The greatest challenge existed in the synthesis of eburnane-type alkaloids is how to effectively control the *cis* C20/C21 relative stereochemistry. Till now, synthetic work that has been successfully achieved with excellent diastereoselectivity was rare[92,93]. In this work, we sought to provide a solution to control the *cis* C20/C21 relative stereochemistry in the synthesis of eburnane-type alkaloids based on our developed method. As shown in Fig. 7a, our synthesis towards eburnane-type alkaloids started with Boc protection of **5a** to get **53** in almost quantitative yield. Ketone in **53** was transformed to alkene by enol-triflation and Pd(PPh$_3$)$_4$-catalyzed reduction, affording **54** in 67% yield over two steps. Similar N–O cleavage with allyl bromide proceeded well and delivered ester **55** in 89% yield when allyl bromide was used. RCM reaction followed by deprotection of Boc generated cyclization product **56**, which was then converted to **57** in 70% yield by stereoselective alkylation[94]. We surmised that the perfect control of *cis* C20/C21 relative stereochemistry might be rationalized in terms of electronic interactions between *trans*- and *cis*-dianionic intermediate **A** and **A'**. Since *cis*-dianionic intermediate **A'** was favored because of the charge repulsion between indole anion and ester enolate in *trans*-dianionic intermediate **A** (See Supplementary Fig. 1), electrophilic attack of ethyl iodide from the convex face of **A'** afforded **57** as single stereoisomer. Next, the ester group in **57** was reduced to hydroxy group in **58** with DIBAL-H. When LiAlH$_4$ reduction condition was subjected to **57**, natural product (−)-14,15-dehydro-17-nor-eburnamonine **59** was obtained. Then, the needed CN unit was installed by mesylation of hydroxy group in **58** and cyanation of the resulting OMs group, furnishing the key intermediate **60**. DIBAL-H reduction of **60** afforded the total synthesis of natural products (−)-Δ$^{14}$-isoeburnamine **61** in 46% yield and (+)-Δ$^{14}$-eburnamine **62**[95,96] in 53% yield. PtO$_2$ catalyzed hydrogenation of double bond in **62** generated natural product (+)-eburnamine **63**[97] in 84% yield. **62** could also be converted to natural product (+)-Δ$^{14}$-eburnamonine **64**[96] by Ley oxidation, therefore accomplishing the total synthesis of (+)-Δ$^{14}$-eburnamonine **64**. Hydrogenation of **64** and **61** delivered (−)-eburnamonine **65**[97] and (−)-isoeburnamine **66** respectively. Under acidic condition (TFA/DCM), (−)-isoeburnamine **66** could be transformed to (−)-eburnamenine **67**[98] in 86% yield. Similarly, **61** and **62** could also be transformed to natural product (−)-Δ$^{14}$-eburnamenine **68** under acidic condition. C14/C15 unsaturated bond existed in many eburnane-type natural products, but the total syntheses of this type of natural products were less explored. Our strategy provided a powerful method to construct these natural products. As depicted in Fig. 7b, the key intermediate **60** could be further transformed to other eburnane-type alkaloids. The key intermediate **60** was reduced by DIBAL-H followed by different

work-up conditions. HCl in MeOH condition afforded the total synthesis of natural product (+)-$\Delta^{14}$-O-methyl-eburnamine **69** and the total synthesis of natural product (−)-$\Delta^{14}$-O-methyl-isoeburnamine **70**[96]. HCl in EtOH condition gave rise to ethoxy-substituted products **73** and **74**. Natural products (+)-O-methyl-eburnamine **71**[97], (−)-O-methyl-iso-eburnamine **72**[97], (+)-O-ethyl-eburnamine **75**[97] and (−)-O-ethyl-iso-eburnamine **76**[97] could be obtained respectively by reduction of the double bond in **69, 70, 73** and **74**. Oxidation of **58** followed

by treatment of the resultant aldehyde immediate with TMSCN delivered **77**, which was then converted to **78** by cyano-group hydrolysis and intramolecular amidation. Ultimately, the total synthesis of natural product (+)-eburnamaline **79**[98] was achieved by a two-step reduction of amide and alkene in **78** with LiAlH$_4$ and PtO$_2$/H$_2$. So far, total synthesis of 16 eburnane-type natural products have been accomplished through our developed asymmetric cycloaddition method.

**Fig. 6 | Total synthesis from product 3. a** (a) BnBr, MeCN, 25 °C, then DABCO, reflux, then (Boc)$_2$O, DMAP, Et$_3$N, 25 °C; (b) DIBAL-H, Toluene, −78 °C; (c) ClPh$_3$PCH$_2$OCH$_3$, $^t$BuOK, THF, 0–25 °C, then HCO$_2$H, DCM, 25 °C; (d) Pd(OH)$_2$/C, H$_2$, MeOH, 25 °C; (e) TFA, DCM, 25 °C. **b** (a) Allyl bromide, MeCN, 25 °C, then DABCO, reflux, then (Boc)$_2$O, DMAP, Et$_3$N, 25 °C; (b) DIBAL-H, Toluene, −78 °C; (c) Ph$_3$PMe•Br, $^t$BuOK, THF, 25 °C; (d) Grubbs II, Toluene, 80 °C; (e) Pd/C, H$_2$, EtOH, 25 °C; (f) TFA, DCM, 25 °C. **c** (a) (Z)-1-Bromo-2-iodo-2-butene, MeCN, 25 °C, then DABCO, reflux, then (Boc)$_2$O, DMAP, Et$_3$N, 25 °C; (b) DIBAL-H, Toluene, −78 °C; (c) MsCl, Et$_3$N, DCM, 0–25 °C, then TMSCN, TBAF, MeCN, 25 °C; (d) MeMgBr, Et$_2$O, 0–25 °C; (e) Pd(PPh$_3$)$_4$, $^t$BuOK, THF, reflux, then silica gel, Toluene, reflux; (f) DIBAL-H, Toluene, −78 °C; (g) (MeO)$_2$P(O)CH$_2$CO$_2$Me, NaH, THF, 25 °C; (h) TFA, DCM, 0–25 °C; (i) Ni(COD)$_2$, Et$_3$N, Et$_3$SiH, MeCN, 25 °C; (j) Crabtree's catalyst, H$_2$, DCM, 25 °C; (k) LiAlH$_4$, THF, 0–25 °C; (l) LDA, THF, −78 °C, then HCO$_2$Me, −78–25 °C; (m) TMSCHN$_2$, DIPEA, MeCN, MeOH, 25 °C; (n) PtO$_2$, H$_2$, MeOH, 25 °C. **d** (a) 2,3-Dibromopropene, MeCN, 25 °C, then DABCO, reflux, then (Boc)$_2$O, DMAP, Et$_3$N, 25 °C; (b) DIBAL-H, Toluene, −78 °C; (c) (MeO)$_2$P(O)CH$_2$CO$_2$Me, NaH, THF, 25 °C; (d) TFA, DCM, 25 °C; (e) AIBN, $^n$Bu$_3$SnH, Toluene, reflux; (f) (Boc)$_2$O, DMAP, Et$_3$N, DCM, 25 °C; (g) DIBAL-H, Toluene, −78 °C; (h) Allylmagnesium bromide, THF, 0 °C; (i) Hoveyda–Grubbs II, DCM, reflux; (j) Pd/C, H$_2$, MeOH, 25 °C; (k) TFA, DCM, 25 °C; (l) DCC, DMSO, Cl$_2$CHCO$_2$H, 35 °C; (m) DCC, Cl$_2$CHCO$_2$H, DMSO, 35 °C, then TFA, DCM, 25 °C. **e** (a) MeI, MeCN, 25 °C, then DABCO, reflux, then (Boc)$_2$O, DMAP, Et$_3$N, reflux; (b) DIBAL-H, Toluene, −78 °C; (c) (MeO)$_2$POCH$_2$CO$_2$Me, NaH, THF, 25 °C; (d) PtO$_2$, H$_2$, MeOH, 25 °C; (e) DIBAL-H, Toluene, −78 °C, then TFA, DCM, 0–25 °C; (f) DIBAL-H, Toluene, −78 °C, then TFA, H$_2$O, THF, 0–25 °C.

Given the prominence of chiral 1,1-disubstituted THβCs with C1-quaternary stereocenter, we then attempted to prove the power of this method in total synthesis of 1,1-disubstituted THβC containing indole alkaloid from chiral tetrahydro-β-carboline fused isoxazolidine product **7b** with quaternary stereocenter. As depicted in Fig. 7c, compound **80** was obtained via reduction of ester group in **7b** followed by TBDPS protection, which was converted to primary alcohol by SmI$_2$/MeOH promoted N–O bond cleavage, then selective alkylation of secondary amine generated **81**. Mesylation of the primary hydroxyl group and replacement of the resulting mesylate by cyano group followed by DIBAL-H reduction and PDC oxidation generated amide **82** in 51% yield over four steps. Next, conversion of **82** into the α,β-unsaturated amide **83** was performed by a two-step synthetic sequence including the introduction of phenyl-selenyl group and subsequent oxidative cleavage to generate the double bond. A Ni(COD)$_2$-promoted reductive cyclization gave the corresponding product **84**. By removing TBDPS group of **84** and reduction of amide, the total synthesis of (−)-arbornamine **85**[99] was completed.

Having validated our catalytic asymmetric 1,3-dipolar cycloaddition method in total synthesis of various natural products, enantiodivergent synthesis was next pursued. In the case of eburnane-type indole alkaloids, which were characterized by naturally occurrence of enantiomeric pairs, both enantiomers were isolated[97]. To achieve the enantiodivergent total synthesis of eburnane alkaloids, we sought to apply our developed methods to the synthesis of the full complement of stereoisomeric eburnane indole natural products from the same precursors by adjustment of the stereochemistry of CPA catalyst. To this end, we used **ent-5a** as a starting material to synthesize corresponding enantiomeric natural products (−)-eburnamine **ent-63**, (+)-eburnamonine **ent-65**, (+)-isoeburnamine **ent-66**, (+)-eburnamenine **ent-67**, (−)-O-methyl-eburnamine **ent-71**, (+)-O-methyl-isoeburnamine **ent-72**, (−)-O-ethyl-eburnamine **ent-75** and (+)-O-ethyl-isoeburnamine **ent-76** according to the same synthetic routes with similar efficiency and enantioselectivity (Fig. 8). The realization of eburnane alkaloids synthesis with antipodal enantiomeric congeners illustrated that our developed method can readily be applied to the design of enantiodivergent transformations.

## Discussion

In summary, we have developed the catalytic asymmetric 1,3-dipolar cycloaddition of 3,4-dihydro-β-carboline-2-oxide with vinyl ether, providing a highly efficient protocol toward three types of chiral tetrahydro-β-carboline fused isoxazolidine generating two stereocenters, three continuous stereocenters, and quaternary stereocenters in high yields with excellent diastereoselectivities and enantioselectivities through varying easily accessible vinyl ether substrates. The *endo/exo* selectivity of our 1,3-dipolar cycloaddition differed from traditional nitrone chemistry. The applicability of our developed method has been realized to collective syntheses of 40 THβC-type indole alkaloid natural products belonging to five families. The method also enabled enantiodivergent total synthesis, thus facilitating the synthesis efficiency. In future, the generality and modularity of this

asymmetric cycloaddition method are expected to be shown in total synthesis of other indole alkaloids due to the valuable tetrahydro-β-carboline skeleton.

## Methods
### General methods

Unless otherwise mentioned, all reagents were purchased from commercial suppliers without further purification. Solvent purification was conducted according to Purification of Laboratory Chemicals (Peerrin, D. D.; Armarego, W. L. and Perrins, D. R., Pergamon Press: Oxford, 1980). Reactions were monitored using Merck Kieselgel 60F$_{254}$ aluminum plates. TLC was visualized by UV fluorescence (254 nm) then one of the following: KMnO$_4$, phosphomolybdic acid, ninhydrin, p-anisaldehyde, vanillin. If not specially mentioned, flash column chromatography was performed using Yantai xinnuo Chemicals (China) (particle size 0.040–0.063 mm). NMR spectra were recorded on JEOL 400 instruments or Bruker Avance NEO 400 and calibrated by using residual undeuterated chloroform-d ($\delta$ $^1$H = 7.26 ppm, $\delta$ $^{13}$C = 77.0 ppm) and DMSO-d$_6$ ($\delta$ $^1$H = 2.55 ppm, $\delta$ $^{13}$C = 39.5 ppm) as internal references. The following abbreviations were used to explain the multiplicities: s = singlet, d = doublet, t = triplet, q = quartet, b = broad, td = triple doublet, dt = double triplet, dq = double quartet, m = multiplet. Infrared (IR) spectra were recorded on an iCAN 9-T FT-IR spectrometer. High-resolution mass spectra (HRMS) were recorded on a Thermo Fisher Q Exactive Orbitrap mass spectrometer using ESI (electrosprayionization) as ionization method.

### General procedure for the synthesis of 3

To a stirred solution of 3,4-dihydro-β-carboline 2-oxide **1** (0.20 mmol, 1.0 equiv), (S)-phosphoric acid **4b** (0.02 mmol, 0.1 equiv) and 3 Å molecular sieves (300 mg) in CHCl$_3$ (2.0 mL) was added ethyl vinyl ether **2a** (0.40 mmol, 2.0 equiv) at −60 °C. The reaction was stirred at −60 °C until TLC indicated that the 3,4-dihydro-β-carboline 2-oxide disappeared. The reaction mixture was directly charged to column chromatography on silica gel (petroleum ether: EtOAc, 4:1-2:1) to give the product.

### General procedure for the synthesis of 5

To a stirred solution of 3,4-dihydro-β-carboline 2-oxide **1** (0.20 mmol, 1.0 equiv), (S)-phosphoric acid **4b** (0.02 mmol, 0.1 equiv) and 3 Å molecular sieves (300 mg) in CHCl$_3$ (2.0 mL) was added (E)-4-methoxy-3-buten-2-one **2b** (0.40 mmol, 2.0 equiv) at −60 °C. The reaction was stirred at −60 °C until TLC indicated that the 3,4-dihydro-β-carboline 2-oxide disappeared. The reaction mixture was directly charged to column chromatography on silica gel (DCM: EtOAc, 20:1) to give the product **5**.

### General procedure for the synthesis of 7

To a stirred solution of 1-oxycarbonyl-3,4-dihydro-β-carboline 2-oxide **6** (0.20 mmol, 1.0 equiv), (S)-phosphoric acid **4a** (0.02 mmol, 0.1 equiv) and 3 Å molecular sieves (300 mg) in DCE (2.0 mL) was added ethyl vinyl ether **2a** (0.40 mmol, 2.0 equiv) at −60 °C. The reaction was

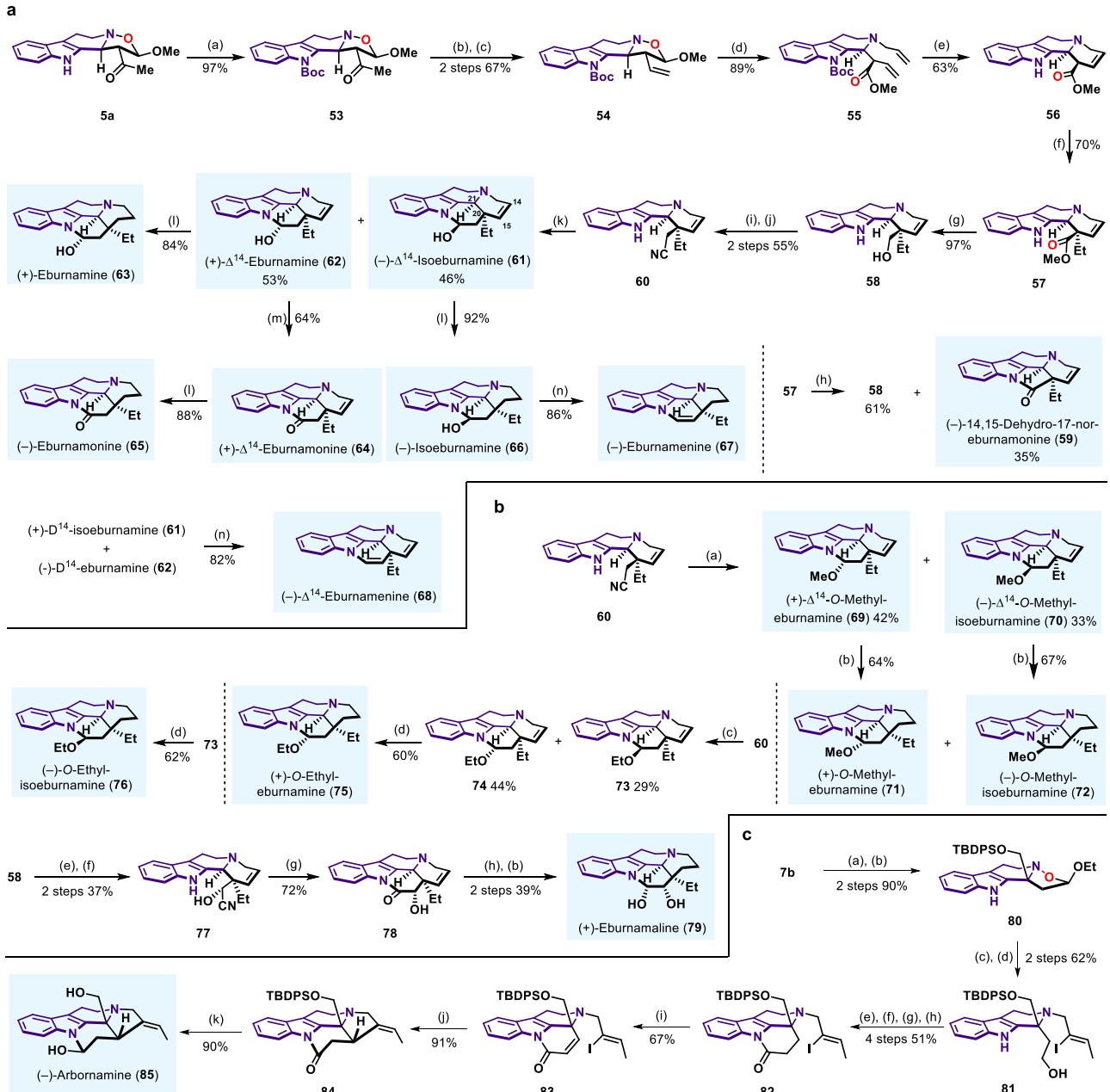

**Fig. 7 | Total synthesis from product 5a and 7b. a** Total syntheses of eburnane-type indole alkaloids. Reagents and conditions: (a) (Boc)₂O, DMAP, Et₃N, DCM, 25 °C; (b) Comins' Reagent, LiHMDS, THF, −78 °C; (c) LiCl, Pd(PPh₃)₄, ⁿBu₃SnH, THF, 25 °C; (d) Allyl bromide, MeCN, 25 °C, then DABCO, reflux; (e) Grubbs II, DCM, reflux, then TFA, DCM, 0–25 °C; (f) LDA, HMPA, −78 °C, then EtI, THF, −40 °C; (g) DIBAL-H, Toluene, 0 °C; (h) LiAlH₄, THF, 0–25 °C; (i) MsCl, Et₃N, DCM, 0–25 °C; (j) TMSCN, TBAF, MeCN, 25 °C; (k) DIBAL-H, Toluene, −78 °C; (l) PtO₂, H₂, EtOH, 25 °C; (m) TPAP, NMO, DCM, 0–25 °C; (n) TFA, DCM, 0–25 °C. **b** Total syntheses of eburnane-type indole alkaloids. Reagents and conditions: (a) DIBAL-H, Toluene,

−78 °C then HCl in MeOH, 0–25 °C; (b) PtO₂, H₂, MeOH, 25 °C; (c) DIBAL-H, Toluene, −78 °C then HCl in EtOH, 0–25 °C; (d) PtO₂, H₂, EtOH, 25 °C; (e) SO₃•Py, DMSO, Et₃N, 0 °C-25 °C; (f) TMSCN, AlCl₃, CHCl₃, 0–25 °C; (g) conc. HCl, MeOH, 80 °C; (h) LiAlH₄, THF. 0–25 °C. **c** Total syntheses of (−)-arbornamine. Reagents and conditions: (a) LiAlH₄, THF, 0–25 °C; (b) TBDPSCl, Imidazole, DMF, 25 °C; (c) SmI₂, MeOH, THF, 25 °C; (d) (Z)-1-Bromo-2-iodo-2-butene, K₂CO₃, MeCN, 25 °C; (e) MsCl, Et₃N, DCM, 0–25 °C; (f) TMSCN, TBAF, MeCN, 25 °C; (g) DIBAL-H, Toluene, −78 °C; (h) PDC, DCM, 25 °C; (i) LiHMDS, THF, PhSeBr, −78 °C, then aq. NH₄Cl, H₂O₂, 0 °C; (j) Ni(COD)₂, Et₃N, Et₃SiH, MeCN, 25 °C; (k) TBAF, THF, 0–25 °C, then LiAlH₄, 0 °C.

stirred at −60 °C until TLC indicated that the 1-oxycarbonyl-3,4-dihy-dro-β-carboline 2-oxide disappeared. The reaction mixture was directly charged to column chromatography on silica gel (petroleum ether: EtOAc, 4:1) to give the product **7**.

### Computational methods
All calculations were carried out with the Gaussian 16 package[100]. Geometry optimization and energy calculations were performed with B3LYP-D3[101]. The 6-31G(d) basis set[102–104] was used for all atoms. Frequency analysis was conducted at the same level of theory to verify that the stationary points are minima or saddle points. Single point energies were calculated at the B3LYP-D3/def2TZVP level using SMD solvation model[105]. The frontier molecular orbitals (FMOs) and their energies were computed at the HF/6-31G(d) level using the B3LYP-D3/6-31G(d) geometries. The CYLview[106] software was employed for visualizations.

### Data availability
Details about materials and methods, experimental procedures, mechanistic studies, characterization data, and NMR spectra are

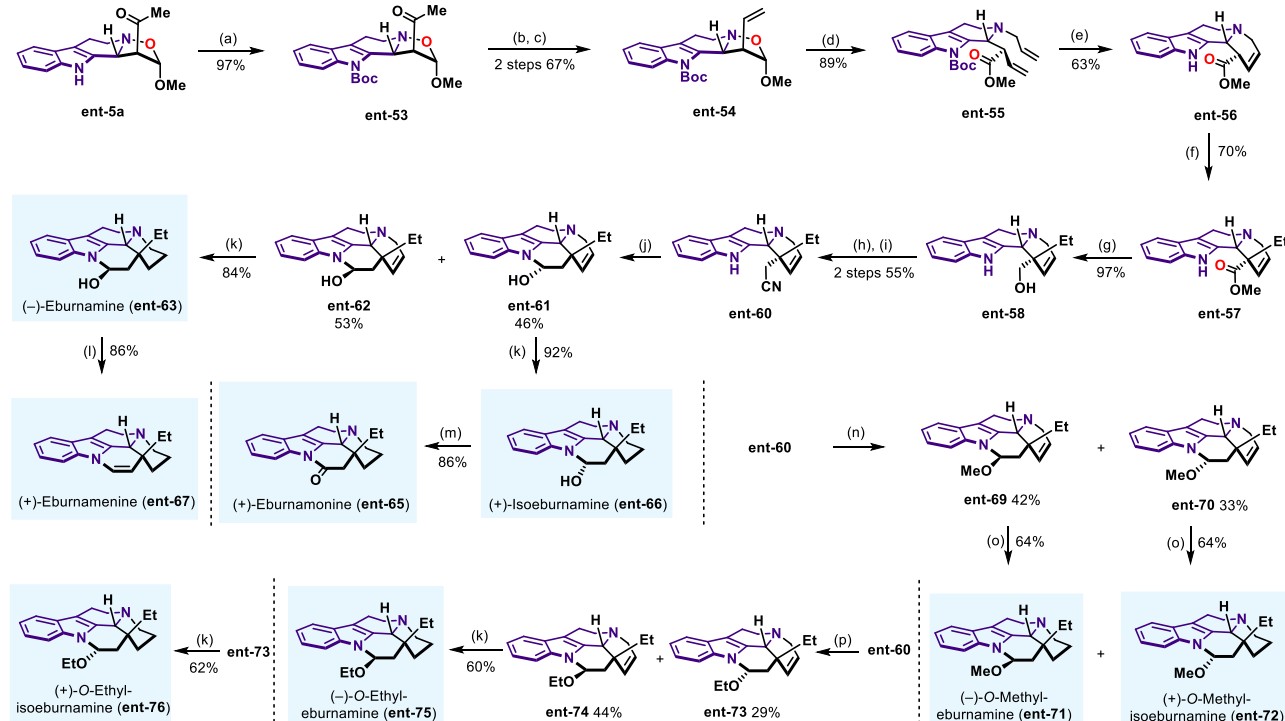

**Fig. 8 | Enantiodivergent total syntheses of eburnane-type alkaloids.** Reaction conditions: (a) (Boc)₂O, DMAP, Et₃N, 25 °C; (b) Comins' Reagent, LiHMDS, THF, −78 °C; (c) LiCl, Pd(PPh₃)₄, ⁿBu₃SnH, THF, 25 °C; (d) Allyl bromide, MeCN, 25 °C, then DABCO, reflux; (e) Grubbs II, DCM, reflux, then TFA, DCM, 0–25 °C; (f) LDA, HMPA, −78 °C, then EtI, THF, −40 °C; (g) DIBAL-H, Toluene, 0 °C; (h) MsCl, Et₃N, DCM, 0–25 °C; (i) TMSCN, TBAF, MeCN, 25 °C; (j) DIBAL-H, Toluene, −78 °C; (k) PtO₂, H₂, EtOH, 25 °C; (l) TFA, DCM, 0–25 °C; (m) TPAP, NMO, DCM, 0–25 °C; (n) DIBAL-H, Toluene, −78 °C then HCl in MeOH, 0–25 °C; (o) PtO₂, H₂, MeOH, 25 °C; (p) DIBAL-H, Toluene, −78 °C then HCl in EtOH, 0–25 °C.

available in the Supplementary Information. Additional data are available from the corresponding author upon request. Crystallographic data are available from the Cambridge Crystallographic Data Centre with the following codes: **3g** (CCDC 2165190), **5d** (CCDC 2165189), **7b** (CCDC 2165191) and **7h** (CCDC 2289028). These data can be obtained free of charge from www.ccdc.cam.ac.uk/data_request/cif. Source data are provided in this paper. Source data are provided with this paper.

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

## Acknowledgements

We thank Qingdao Marine Science and Technology Center (No. 2022QNLM030003-2), the Fundamental Research Funds for the Central Universities, Taishan Scholar Program of Shandong Province (No. tsqn202103152), National Natural Science Foundation of China (No. 22201270), Natural Science Foundation of Shandong Province (No. ZR2021QB033), the National Key Research and Development Program of China (No. 2022YFC2804400). National Natural Science Foundation of China (22371256, 22138011), and the Fundamental Research Funds for the Provincial Universities of Zhejiang (RF-C2022006) for financial support.

## Author contributions

Y.W. designed the study, oversaw the research, and wrote the paper. X.T. and T.X. conducted the experiments with help from X.Z., T.L., F.L., R.P., and P.S. J.G. carried out the DFT calculations. Y.Y. supervised the DFT calculations. X.T., T.X., and J.G. contributed equally to this work. All authors contributed to the analysis and interpretation of the data.

## Competing interests

The authors declare no competing interests.
