## [Peer Review File · Nature Communications]

Catalytic Enantioselective Nitrene Cycloadditions Enabling Collective Syntheses of Indole AlkaloidsREVIEWER COMMENTS

Reviewer #1 (Remarks to the Author):

The paper described a novel chiral phosphoric acid-catalyzed asymmetric 1,3-dipolar cycloaddition of 3,4-dihydro- β -carboline-2-oxide type nitron, leading to the synthesis of chiral tetrahydro- β -carbolines with multiple continuous chiral centers and quaternary chiral centers.

This method demonstrates a distinct endo/exo selectivity, differentiating it from traditional nitron chemistry.

The utility of this synthesis is further underscored by its application in the total syntheses of 40 TH β C-type indole alkaloid natural products, including the first total syntheses of 7 natural products with divergent stereochemistry and varied architectures.

While the yields are moderate, the high degree of stereocontrol in the asymmetric reaction with nitron is commendable. Successfully leading to the synthesis of 7 natural products highlights the practicality and relevance of the synthesized compounds.

However, I suggest that the reaction mechanism could be further elucidated using molecular orbital calculations. This would provide a deeper understanding of the underlying principles governing the observed stereochemical outcomes.

In conclusion, considering the novel approach, the successful application in synthesizing a range of complex natural products, and the potential for improved understanding through additional mechanistic studies, I recommend that this paper, with revisions addressing the above comment, is suitable for publication in Nature Communications.

Reviewer #2 (Remarks to the Author):

Natural products have long been important source of new drug discoveries. Methodology development allowing the synthesis of a large collection of natural products remains scarce. The manuscript by Tian et al. outlines a collective total synthesis of 40 indole alkaloids that leverages an enantioselective 1,3-dipolar cycloaddition of 3,4-dihydro- β -carboline-2-oxide type nitron with different vinyl ethers. This powerful methodology enabled access to three types of chiral tetrahydro- β -carbolines bearing continuous multi-chiral centers and quaternary chiral centers. Moreover, this nitron cycloaddition is featured with high yields, excellent diastereoselectivities and enantioselectivities (80% to 99% ee) and broad substrate scopes. This work therefore, introduces a creative strategy for the synthesis of tetrahydro- β -carboline indole alkaloids. Furthermore, the detailed exploration of cycloaddition reaction conditions will help to guide future optimization of this kind of reaction in other studies.

This is a very well written manuscript that is easy to follow. The only rather small weakness of the paper is that the mechanism study of 1,3-dipolar cycloaddition which is not convincing. This hopefully will be something that can be addressed by adding density functional theory (DFT) calculations. Overall, this work is likely to be of interest to those working on the field of organic synthesis and would have potentially high impact. In general, I have no major issues with these syntheses. I only have some minor suggests to help make some aspects of the study clearer to a general reader. I would suggest publishing this manuscript after minor revision.

Comments:

1. The endo/exo selectivity of 1,3-dipolar cycloaddition in this manuscript is definitely differed from Yamamoto's works (Angew. Chem. Int. Ed. 47,2411-2413(2008)). The authors

explained this difference by repulsive effect and steric hindrance between the function group of the substrates and the catalyst. The density functional theory (DFT) calculations would be clearer.

2. The authors might need to explain the spectral differences of synthetic sample and the reported ones (NP 32 and 41). Have they tried titrations? And they should align the abscissa of these spectra to the same. Furthermore, for the ¹H and ¹³C NMR data comparison, the errors between Natural and Synthetic are recommended to list and make it more clear to readers.

3. The description, found in the discussion, "few methodologies have been widely applied to the total synthesis of TH β C-type natural products and bioactive compounds" does not seem appropriate as written, there are 7 references cited.

4. A recent review related to this manuscript (Green Synth. Catal. 2022,3,25-39) could be cited.

5. Check the typing mistakes (minus signs, DIBAL-H etc..).

Reviewer #3 (Remarks to the Author):

Wang and co-workers describe in this manuscript enantioselective [3+2] cycloaddition reaction of nitrones derived from tetrahydrocarboline skeleton and alkenes using chiral phosphoric acid. Corresponding cycloadducts were obtained in good yields and with high to excellent enantioselectivities. The authors successfully applied the cycloaddition reaction to the total synthesis of tetrahydro-b-carboline derivatives.

Although this manuscript contains interesting experimental results, the results has not reached the high standard of JACS and this reviewer cannot warrant publication in JACS in the present form. Although chiral phosphoric acid-catalyzed [3+2] cycloaddition reaction of nitrones and alkenes were already reported (ref. 61 and 62), the results described in this manuscript will attract much attention of organic chemists, and this reviewer recommends publication of the manuscript in Nat. Commun. after addressing following issues.

(1) It is recommended to try DFT calculations for the transition state model.

(2) Although authors cited chiral phosphoric acid-catalyzed [3+2] cycloaddition reaction of nitrones and alkenes (ref. 61 and 62) in page 12, it should be mentioned in the introductory section clearly.

(3) Several review articles and/or seminal papers of chiral phosphoric acid should be cited. For examples of reviews: Akiyama, T. Chem. Rev. 2007, 107, 5744-5758. Terada, M. Synthesis 2010, 1929-1982. Parmar, D.; Sugiono, E.; Raja, S.; Rueping, M. Chem. Rev. 2014, 114, 9047-9153. Parmar, D.; Sugiono, E.; Raja, S.; Rueping, M. Chem. Rev. 2017, 117, 10608-10620. Merad, J.; Lalli, G.; Bernadat, G.; Maur, J.; Masson, G. Chem. Eur. J. 2018, 24, 3925. Maji, R.; Mallojjala, S. C.; Wheeler, S. E., Chem. Soc. Rev. 2018, 47, 1142. Li, X.; Song, Q. Chin. Chem. Lett. 2018, 29, 1181-1192. Rahman, A.; Lin, X. Org. Biomol. Chem. 2018, 16, 4753-4777. Liu, W.; Yang, X. Asian J. Org. Chem. 2021, 10, 692-710. Maji, R.; Mallojjalaa, S. C.; Wheeler, S. E. Chem. Soc. Rev. 2018, 47, 1142-1158. For seminal works: Akiyama, T.; Itoh, J.; Yokota, K.; Fuchibe, K. Angew. Chem. Int. Ed. 2004, 43, 1566-1568. Uruguchi, D.; Terada, M. J. Am. Chem. Soc. 2004, 126, 5356-5357.

Answers to Comments of Referee 1

Reviewer 1's general comments: The paper described a novel chiral phosphoric acid-catalyzed asymmetric 1,3-dipolar cycloaddition of 3,4-dihydro- β -carboline-2-oxide type nitrene, leading to the synthesis of chiral tetrahydro- β -carbolines with multiple continuous chiral centers and quaternary chiral centers. This method demonstrates a distinct endo/exo selectivity, differentiating it from traditional nitrene chemistry. The utility of this synthesis is further underscored by its application in the total syntheses of 40 TH β C-type indole alkaloid natural products, including the first total syntheses of 7 natural products with divergent stereochemistry and varied architectures. While the yields are moderate, the high degree of stereocontrol in the asymmetric reaction with nitrene is commendable. Successfully leading to the synthesis of 7 natural products highlights the practicality and relevance of the synthesized compounds.

Response: Thank you very much for the professional comments and recommendation.

(1) **Referee 1 wrote:** However, I suggest that the reaction mechanism could be further elucidated using molecular orbital calculations. This would provide a deeper understanding of the underlying principles governing the observed stereochemical outcomes.

Response: To further understand the reaction mechanism, DFT calculations for the transition state model have been performed, and have been added in the revised manuscript.

(2) **Referee 1 wrote:** In conclusion, considering the novel approach, the successful application in synthesizing a range of complex natural products, and the potential for improved understanding through additional mechanistic studies, I recommend that this paper, with revisions addressing the above comment, is suitable for publication in Nature Communications..

Response: . Thank you very much for the recommendation.

Answers to Comments of Referee 2

Reviewer 2's general comments: Natural products have long been important source of new drug discoveries. Methodology development allowing the synthesis of a large collection of natural products remains scarce. The manuscript by Tian et al. outlines a collective total synthesis of 40 indole alkaloids that leverages an enantioselective 1,3-dipolar cycloaddition of 3,4-dihydro- β -carboline-2-oxide type nitrene with different vinyl ethers. This powerful methodology enabled access to three types of chiral tetrahydro- β -carbolines bearing continuous multi-chiral centers and quaternary chiral centers. Moreover, this nitrene cycloaddition is featured with high yields, excellent diastereoselectivities and enantioselectivities (80% to 99% ee) and broad substrate scopes. This work therefore, introduces a creative strategy for the synthesis of tetrahydro- β -carboline indole alkaloids. Furthermore, the detailed exploration of cycloaddition reaction conditions will help to guide future optimization of this kind of reaction in other studies. This is a very well written manuscript that is easy to follow. The only rather small weakness of the paper is that the mechanism study of 1,3-dipolar cycloaddition which is not convincing. This hopefully will be something that can be addressed by adding density functional theory (DFT) calculations. Overall, this work is likely to be of interest to those working on the field of organic synthesis and would have potentially high impact. In general, I have no major issues with these syntheses. I only have some minor suggests to help make some aspects of the study clearer to a

general reader. I would suggest publishing this manuscript after minor revision.

Response: Thank you very much for the professional comments and recommendation. Followings please find our replies to all your concerns.

(1) **Referee 2 wrote:** The endo/exo selectivity of 1,3-dipolar cycloaddition in this manuscript is definitely differed from Yamamoto's works (Angew. Chem. Int. Ed. 47,2411-2413(2008)). The authors explained this difference by repulsive effect and steric hindrance between the function group of the substrates and the catalyst. The density functional theory (DFT) calculations would be clearer.

Response: DFT calculations for the transition state model have been added in the revised manuscript.

(2) **Referee 2 wrote:** The authors might need to explain the spectral differences of synthetic sample and the reported ones (NP 32 and 41). Have they tried titrations? And they should align the abscissa of these spectra to the same. Furthermore, for the ¹H and ¹³C NMR data comparison, the errors between Natural and Synthetic are recommended to list and make it more clear to readers.

Response: The solvent used for NMR spectroscopy was CDCl₃. The spectral differences of synthetic sample and the reported ones might be explained by the different PH conditions of deuterated solvent. CDCl₃ pretreated with K₂CO₃ was used for NMR spectroscopy, and ¹H data had only slight differences of synthetic sample and the reported ones while ¹³C data was identical. For the ¹H and ¹³C NMR data comparison, the errors between natural and synthetic had been listed in the revised supporting information.

(3) **Referee 2 wrote:** The description, found in the discussion, "few methodologies have been widely applied to the total synthesis of TH β C-type natural products and bioactive compounds" does not seem appropriate as written, there are 7 references cited.

Response: Revised.

(4) **Referee 2 wrote:** A recent review related to this manuscript (Green Synth. Catal. 2022,3,25-39) could be cited.

Response: The reference was cited in the revised manuscript (Ref 54).

(5) **Referee 2 wrote:** .Check the typing mistakes (minus signs, DIBAL-H etc..).

Response: We have revised all minus signs in the manuscript and supporting information. And we checked other mistakes, which have been marked in the revised manuscript and supporting information.

Answers to Comments of Referee 3

Reviewer 3's general comments: Wang and co-workers describe in this manuscript enantioselective [3+2] cycloaddition reaction of nitrones derived from tetrahydrocarboline skeleton and alkenes using chiral phosphoric acid. Corresponding cycloadducts were obtained in good yields and with high to excellent enantioselectivities. The authors successfully applied the cycloaddition reaction to the total synthesis of tetrahydro-b-carboline derivatives. Although this manuscript contains interesting experimental results, the results has not reached the high standard of JACS and this reviewer cannot warrant publication in JACS in the present form. Although chiral phosphoric acid-catalyzed [3+2] cycloaddition reaction of nitrones and alkenes were already reported (ref. 61 and 62), the results described in this manuscript will attract much attention of organic chemists, and this reviewer recommends publication of the manuscript in Nat. Commun. after addressing

following issues.

Response: Thank you very much for the professional comments and recommendation. Followings please find our replies to all your concerns.

(1) Referee 3 wrote: It is recommended to try DFT calculations for the transition state model.

Response: DFT calculations for the transition state model have been added in the revised manuscript.

(2) Referee 3 wrote: Although authors cited chiral phosphoric acid-catalyzed [3+2] cycloaddition reaction of nitrones and alkenes (ref. 61 and 62) in page 12, it should be mentioned in the introductory section clearly.

Response: Ref 61 and 62 were cited in the introductory section (Page 5).

(3) Referee 3 wrote: Several review articles and/or seminal papers of chiral phosphoric acid should be cited. For examples of reviews: Akiyama, T. Chem. Rev. 2007, 107, 5744-5758. Terada, M. Synthesis 2010, 1929-1982. Parmar, D.; Sugiono, E.; Raja, S.; Rueping, M. Chem. Rev. 2014, 114, 9047–9153. Parmar, D.; Sugiono, E.; Raja, S.; Rueping, M. Chem. Rev. 2017, 117, 10608–10620. Merad, J.; Lalli, G.; Bernadat, G.; Maur, J.; Masson, G. Chem. Eur. J. 2018, 24, 3925. Maji, R.; Mallojjala, S. C.; Wheeler, S. E., Chem. Soc. Rev. 2018, 47, 1142. Li, X.; Song, Q. Chin. Chem. Lett. 2018, 29, 1181-1192. Rahman, A.; Lin, X. Org. Biomol. Chem. 2018, 16, 4753-4777. Liu, W.; Yang, X. Asian J. Org. Chem. 2021, 10, 692-710. Maji, R.; Mallojjalaa, S. C.; Wheeler, S. E. Chem. Soc. Rev. 2018, 47, 1142-1158. For seminal works: Akiyama, T.; Itoh, J.; Yokota, K.; Fuchibe, K. Angew. Chem. Int. Ed. 2004, 43, 1566-1568. Uruguchi, D.; Terada, M. J. Am. Chem. Soc. 2004, 126, 5356-5357.

Response: The corresponding references were cited in the revised manuscript.

REVIEWERS' COMMENTS

Reviewer #1 (Remarks to the Author):

Since the indicated revisions have been revised, we consider the manuscript to be ready for publication in Nature Communications.

Reviewer #2 (Remarks to the Author):

The authors have addressed my concerns in the revised manuscript, which is recommended for publication.

Reviewer #3 (Remarks to the Author):

The authors addressed all the comments in the revised manuscript. This reviewer recommends publication of the manuscript in Nat. Commun. in the present form.